# Learning environment-specific learning rates

**Jonas Simoens** *, **Tom Verguts, Senne Braem**

Department of Experimental Psychology, Ghent University, Belgium

* Jonas.Simoens@ugent.be

## Abstract

People often have to switch back and forth between different environments that come with different problems and volatilities. While volatile environments require fast learning (i.e., high learning rates), stable environments call for lower learning rates. Previous studies have shown that people adapt their learning rates, but it remains unclear whether they can also learn about environment-specific learning rates, and instantaneously retrieve them when revisiting environments. Here, using optimality simulations and hierarchical Bayesian analyses across three experiments, we show that people can learn to use different learning rates when switching back and forth between two different environments. We even observe a signature of these environment-specific learning rates when the volatility of both environments is suddenly the same. We conclude that humans can flexibly adapt and learn to associate different learning rates to different environments, offering important insights for developing theories of meta-learning and context-specific control.

## Author summary

People constantly have to make decisions, such as what to wear for the day, or which pizza to order at a restaurant. Fortunately, people can learn from past decisions to inform future ones. However, environments may be unstable: The best pizza today is not necessarily the best pizza tomorrow. The chef may have had a bad day, in which case no learning needs to take place, or the restaurant may have changed chefs, in which case learning needs to restart from scratch. For this reason, it pays off to learn the instabilities of different environments: Which pizza is best may change more often in one restaurant than in another (e.g., because chefs change more quickly in one restaurant than in another). Formally, environmental instability determines how strongly one should update the expected value of an option (e.g., a pizza) based on novel information, often referred to as the *learning rate*. We thus investigated if people can learn different learning rates for different environments. We demonstrated that they can: Participants randomly and quickly alternated between a stable and an unstable environment, and they learned to use higher learning rates in the unstable than in the stable environment.

**Data Availability Statement:** All data files and data analysis scripts are available on OSF at https://osf.io/492mg/.

**Funding:** This work was funded by an FWO fellowship awarded to JS (#11K5121N), an FWO project grant awarded to TV and SB (G010319N), and an ERC Starting grant awarded to SB

(European Union's Horizon 2020 research and innovation program, Grant agreement 852570). The funders had no role in study design, data collection and analysis, decision to publish, or preparation of the manuscript.

**Competing interests:** The authors have declared that no competing interests exist.

## Introduction

People often have to track and trace multiple events at the same time, such as watching different kids at a playground. When doing so, it pays to learn the different volatilities (i.e., rates of change) and use them for informing one's decisions. For example, one might know that one kid tends to engage in more risky behaviour, which requires monitoring her more closely. The ability to learn and use this knowledge about different volatilities is often considered an important hallmark of human, intelligent behaviour [1,2]. In reinforcement learning theory, environmental volatility determines to what extent one should update the value of an action when this action yields more or less reward than expected (i.e., when one experiences prediction errors), often referred to as the *learning rate*. The question then becomes if humans can learn about learning rates, or "meta-learn" [3–8], that is, if they can learn about different learning rate requirements for different environments. Here, we set out to test this.

A basic tenet of reinforcement learning theory is that agents aim to repeat the action that will result in the most rewarding outcome [9]. As such, action values are crucial for adaptive decision making. In reinforcement learning models [10], these (lower-level) action values are modelled as $Q^a$, representing the value $Q$ of action $a$. At a higher level, additional parameters (such as learning rate) are required to estimate these $Q$ values. While the lower-level $Q$ values are learned by definition, reinforcement learning models typically assume these higher-level parameters to have fixed settings. However, in recent years, theories of meta-learning have suggested that people can also flexibly adapt their learning rates to specific environments [3–8].

For example, it has been shown that people can flexibly adapt their learning rate to the statistics of reward contingencies in their environment [1,11–14], consistent with what optimality analyses predict [15]. Specifically, people use lower learning rates, on average, when probabilistic reward contingencies are stable (i.e., when prediction errors are likely due to noise and should be ignored) than when reward contingencies are volatile (i.e., when prediction errors are likely to signal changed reward contingencies). An important limiting feature of these studies, however, is that environment volatilities are always clustered in time. That is, participants learn under either volatile or stable conditions in separate phases of the experiment, but never intermixed. This way, learning rate adaptations in separate phases of the experiment could reflect fleeting adaptations to differences in experienced prediction errors, with differences in learning rates between environments being a mere emergent property of behaviour. For example, some studies that examined how learning rates evolve on a trial-by-trial basis, have shown that people show temporarily increased learning rates when suddenly experiencing more prediction errors (i.e., after a reward contingency change) [16,17]. Similarly, since the aforementioned studies contrasted stable environments with volatile environments clustered in time [1,11–14], participants' varying learning rates across environments may reflect local adaptations (i.e., learning within an environment), but do not necessarily demonstrate that participants actually learned about the different learning rates, and to associate them to the different environments (i.e., meta-learning about the environment).

In this study, we aimed to provide first evidence for the idea that people can also learn environment-specific learning rates and flexibly switch back and forth between them. As such, we designed three, preregistered experiments in which participants simultaneously performed two two-armed bandit tasks in two different environments (i.e., casinos), one with stable and one with volatile reward contingencies. Crucially, participants randomly alternated between both environments, made recognisable by the contextual cues, on a trial-by-trial basis, requiring them to actively keep track of how to behave differently in both environments. We hypothesised that, even though participants constantly alternated between both environments, their behaviour would be best explained by a model with two different environment-specific

learning rates, as they would learn, over time, to use a higher learning rate in the volatile than in the stable environment. In our third experiment we also used an unannounced test phase where the two environments had the same volatilities. This allowed us to investigate whether we could detect a signature of different learning rates, evidencing that people learned to associate and retrieve these different learning rates based on environmental cues.

## Results

In all experiments, participants would gamble in two different casinos. In each casino, they had to try and learn which of two slot machines had, on average, the most rewarding outcome. Unannounced to the participants, there was a stable and a volatile casino. In the stable casino, the slot machine most likely to yield reward (i.e., 80% vs. 20%) remained constant for the duration of a block, while in the volatile casino, the slot machine most likely to yield reward (i.e., 90% vs. 10%) switched every 16–24 trials (within that casino). Crucially, participants randomly alternated between both casinos, made recognisable by pictures preceding each trial as well as the location and colour of the slot machines, on a trial-by-trial basis (see Fig 1A). As a result, the number of consecutive trials that participants received the same casino was 1 for 50% of the time, 2 for 25.07% of the time, 3 for 12.37% of the time, 4 for 6.4% of the time, 5 for 3.04% of the time, 6 for 1.49% of the time, and anything over 6 for less than 1% of the time.

In both casinos, the rewards were probabilistic such that the more rewarding slot machine would result in wins on most trials. This probability was slightly higher in the volatile casino such that, on average, an optimal learner could obtain a similar amount of rewards in both casinos. These optimal reward probabilities were determined through reward rate simulations (see Methods for more details), which further established that an optimal learner (behaving according to the Rescorla-Wagner model) would use a different learning rate in each environment to obtain the highest reward rate (see Fig 1B). We hypothesised that, even though participants repeatedly switched back and forth between both environments, participants would learn to use a different, higher learning rate in the volatile than in the stable environment.

### Experiment 1

In Experiment 1 (n = 36; https://osf.io/bku6s), participants were first presented with two blocks of 240 trials each in which participants randomly alternated between the stable and the volatile casino (i.e., learning phase). After training, we included a third block of 240 trials in which participants alternated between two new casinos that were both stable (i.e., control phase) (Fig 1A). The latter served as a manipulation check to see whether people would show a difference in learning rate between the different phases of the experiment (higher learning rates in the learning phase than in the control phase), in line with traditional manipulations of volatility where the different environments are clustered in time [1,11–14], even if no difference in learning rate between the within-block environments would have been observed. However, of main interest was the difference between the two environments in the learning phase, where we hypothesised that participants would use higher learning rates in the volatile compared to the stable casino.

First, we fitted different models to the choice data, to see which model could best explain the data. Specifically, we fitted six variations of the Rescorla-Wagner model [18], using hierarchical Bayesian analysis [19], to investigate how participants learned the values of the different slot machines using the delta learning rule and chose between the available slot machines using the softmax decision rule (see Methods for more details). Most importantly, we compared models with either a shared learning rate across both environments, versus models with different, environment-specific learning rates for both casinos. Furthermore, we compared

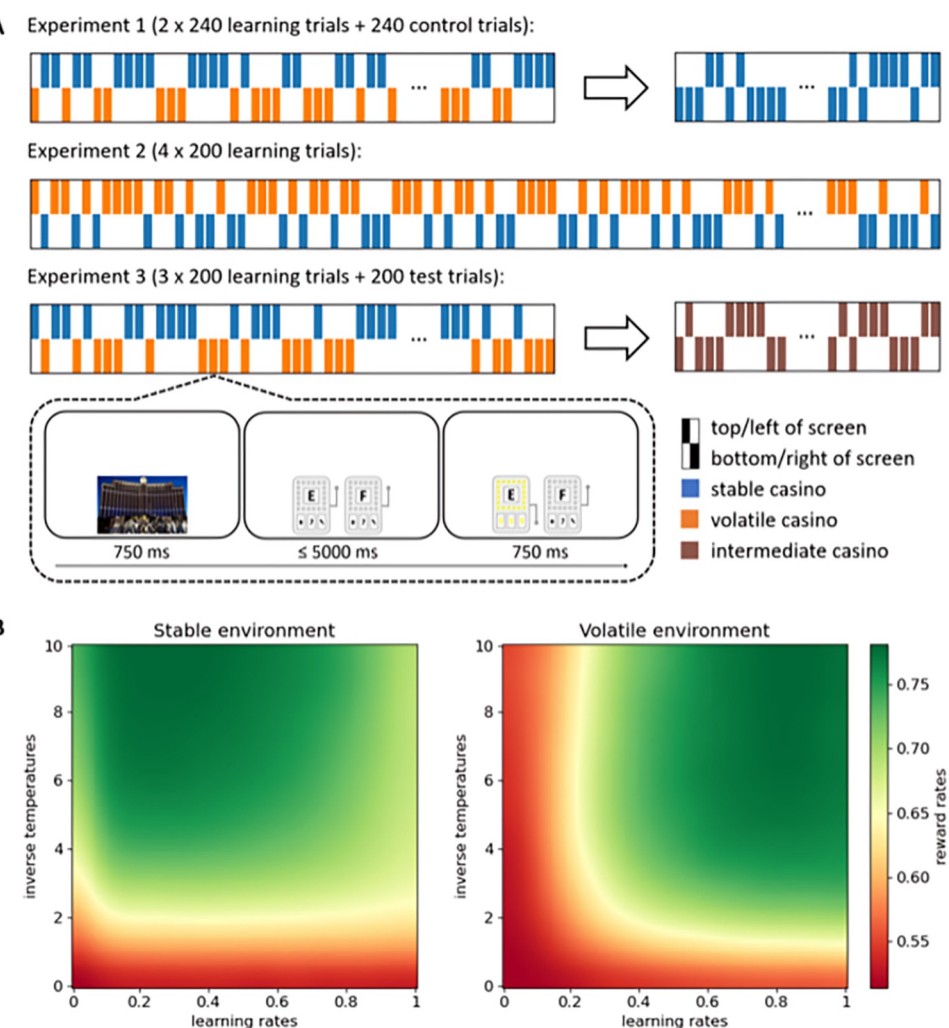

**Fig 1. Experimental design. A:** Experimental procedures. All three experiments started with a learning phase, in which participants randomly alternated on a trial-by-trial basis between the stable and the volatile casino, one of which was always presented on the top of the screen and had blue slot machines, while the other one was always presented on the bottom of the screen and had grey slot machines. The relation between location and the volatility of the casino was counterbalanced across participants. In Experiment 1, the learning phase was followed by a control phase, in which participants randomly alternated on a trial-by-trial basis between two stable casinos, one of which was always presented on the left of the screen and had green slot machines, while the other one was always presented on the right of the screen and had orange slot machines. In Experiment 3, without notifying participants that anything would change, the learning phase was followed by a test phase, in which participants randomly alternated between the same two casinos as in the learning phase, but both of their contingencies were now of intermediate volatility. On each trial, one of the two casinos was first presented on the relevant location for 750 ms. Next, two slot machines were presented in the relevant colour until the participant had chosen one of them, with a response deadline of 5 seconds. Finally, feedback was presented for 750 ms: the lights on the chosen slot machine would start to flicker, if reward was obtained, or remained off, if reward was not obtained. **B:** Reward rate simulation results. Each point represents the (smoothed) reward rate obtained by the simulated Rescorla-Wagner model in the relevant environment and with the relevant parameter settings.

models with separate learning rates after positive and negative reward feedback [20–24], to models with one learning rate for both feedback types. To quantify the alternative hypothesis that learning rate changes merely reflect local adaptations to the experienced prediction errors, we also implemented models with variable learning rates that depend on recently experienced prediction errors [2,16,25,26]. Factorially combining these different kinds of learning rates

**Table 1. Learning phase model comparison.**

| Model | LOOIC | SE | ΔLOOIC | ΔSE |
|---|---|---|---|---|
| Environment-specific dual learning rate model | -6679 | 406 | 0 | 0 |
| Environment-specific single learning rate model | -6724 | 407 | 45 | 15 |
| Environment-specific variable learning rate model | -6743 | 398 | 64 | 42 |
| Non-environment-specific dual learning rate model | -6827 | 413 | 149 | 31 |
| Non-environment-specific single learning rate model | -6861 | 415 | 183 | 33 |
| Non-environment-specific variable learning rate model | -6874 | 411 | 196 | 35 |

*Note.* Models are ranked in descending order according to how well they fit the data. LOOIC refers to a model's approximated expected log pointwise predictive density. Higher values indicate higher out-of-sample predictive fit. SE refers to the standard error of a model's LOOIC. ΔLOOIC refers to the difference between a model's LOOIC and the top ranked model's LOOIC. ΔSE refers to the standard error of the difference between a model's LOOIC and the top ranked model's LOOIC

(i.e., single, dual, and variable) and environment-specific versus non-environment-specific learning rates, we ended up comparing six different models.

Our preregistered hypothesis was that environment-specific models should outperform non-environment-specific models. Moreover, we hypothesized that the difference in environment-specific learning rates should be attributable to their association to the environment, and not a by-product or reaction to more local prediction errors, as the variable learning rate model would predict. Therefore, we further hypothesized that the environment-specific single and dual learning rate models should fit the data better than the non-environment-specific variable learning rate model. However, we were agnostic about whether the environment-specific single or dual learning rate model would perform best.

As can be seen in Table 1, all environment-specific models fitted the data (from the learning phase) better than all non-environment-specific models according to the leave-one-out information criterion (LOOIC) [27]. More specifically, the environment-specific dual learning rate model performed best across all six models.

Since the dual learning rate model fitted the data best, we performed additional reward rate simulations with this model rather than with the single learning rate model (cf. Fig 1) to test whether according to this model, too, an optimal learner would use different learning rates in the two environments. The dual learning rate model has three free parameters, however, which cannot all be visualized in a two-dimensional heatmap. Moreover, in the dual learning rate model reward rate simulations, the interactions between learning rates and inverse temperatures were no different than in the single learning rate model reward rate simulations. Therefore, we only visualised the interactions between positive and negative learning rates, with inverse temperatures fixed to 4, the average observed inverse temperature. However, the results look almost identical with higher, and therefore closer to optimal, inverse temperatures.

As can be seen in Fig 2B, these simulations showed that also an agent with dual learning rates benefits more from higher learning rates in the volatile than in the stable environment. This effect is more pronounced for negative learning rates (i.e., learning from negative prediction errors). More specifically, in the stable environment, lower negative learning rates tend to result in higher reward rates, while positive learning rates have less effect on reward rate. In the volatile environment, higher negative as well as positive learning rates tend to result in higher reward rates, but this effect is still more pronounced for negative than for positive learning rates. This is likely because negative learning rates are more important than positive learning rates when adapting to changed reward contingencies. That is, since one is likely to experience a lot of negative prediction errors immediately following a reward contingency

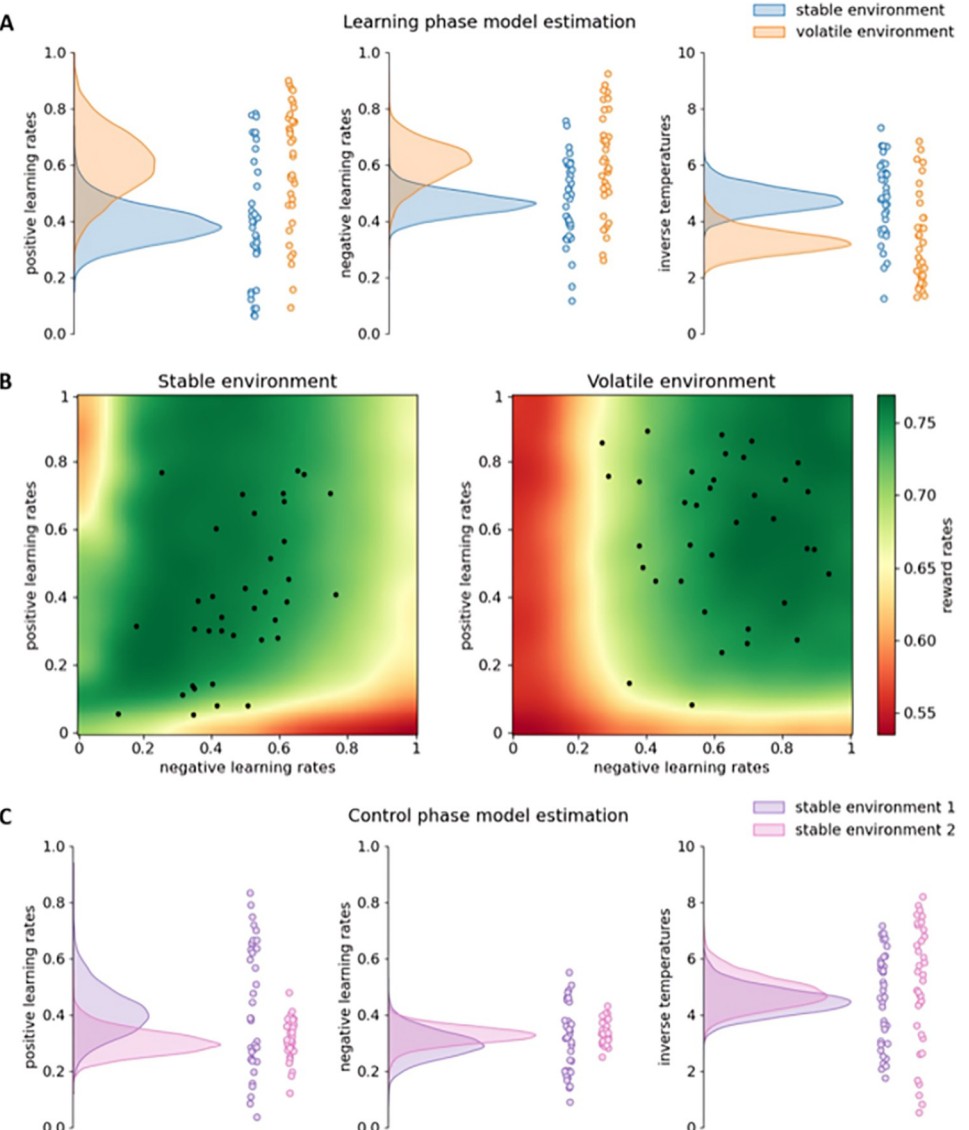

**Fig 2. Results Experiment 1. A:** Model estimation results for the learning phase. The density plots on the left side of each subfigure show the full posterior densities over the means of the group-level distributions of the relevant parameters. The scatter plots on the right side of each subfigure show the means of all individual-level posterior distributions of the relevant parameters. **B:** Reward rate simulation results for the dual learning rate model. Each point represents the (smoothed) reward rate obtained by the simulated dual learning rate model in the relevant environment and with the relevant parameter settings. Each black dot represents a participant's estimated positive and negative learning rate. **C:** Model estimation results for the control phase.

change, a higher negative learning rate will enable one to adapt faster to this change. Crucially, here, reward contingency changes exclusively occurred in the volatile environment.

Next, we computed the posterior probabilities that learning rates were higher in the volatile than in the stable environment. In line with our hypotheses, in the learning phase, the posterior probabilities that positive and negative learning rates were higher in the volatile than in the stable environment were 0.962 and 0.958, respectively (Fig 3A). The posterior probability that inverse temperatures were lower in the volatile than in the stable environment in the learning phase was 0.99. Confirming these results, in the volatile environment participants

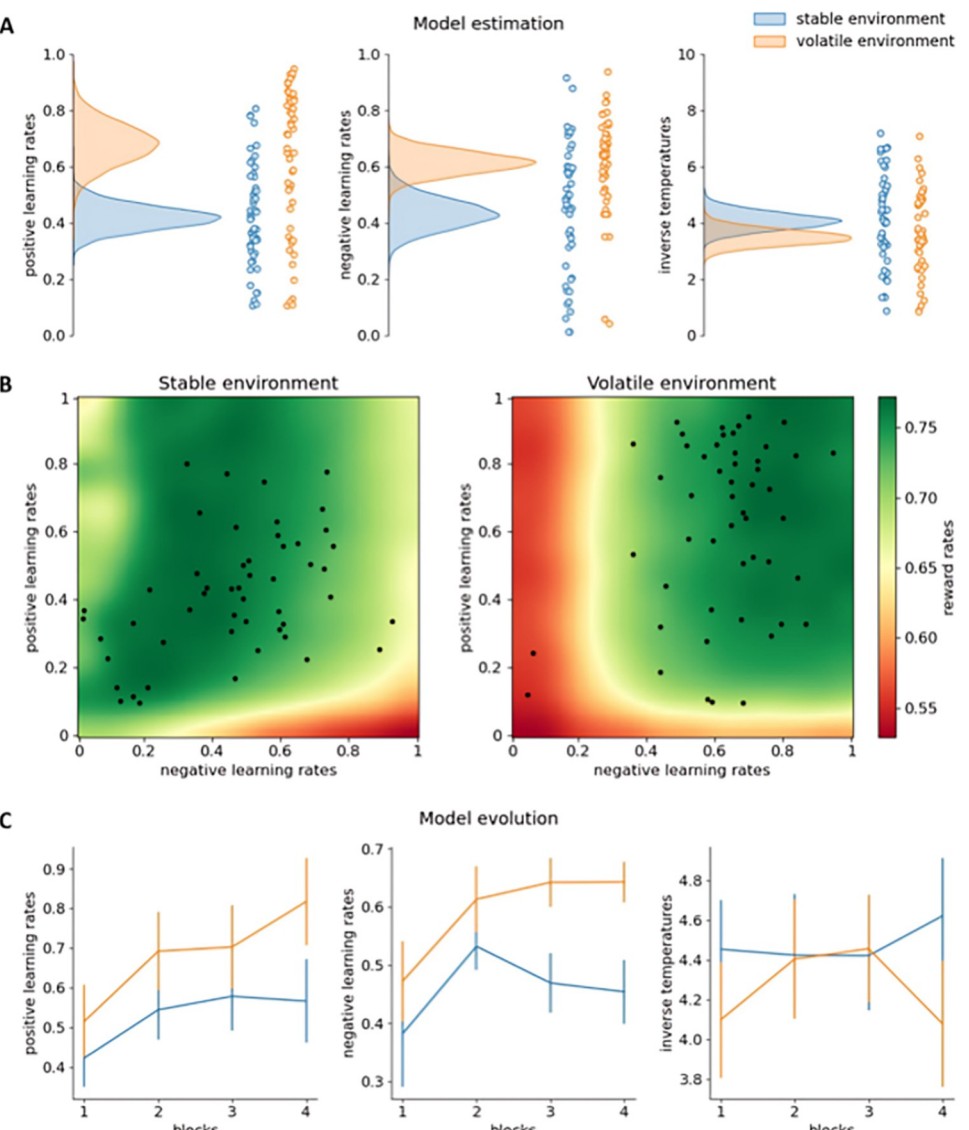

**Fig 3. Results Experiment 2. A:** Model estimation results. The density plots on the left side of each subfigure show the full posterior densities over the means of the group-level distributions of the relevant parameters. The scatter plots on the right side of each subfigure show the means of all individual-level posterior distributions of the relevant parameters. **B:** Reward rate simulation results for the dual learning rate model. Each point represents the (smoothed) reward rate obtained by the simulated dual learning rate model in the relevant environment and with the relevant parameter settings. Each black dot represents a participant's estimated positive and negative learning rate. **C:** Evolution of model parameters over blocks. Each subfigure shows the means and standard deviations of the posterior densities over the means of the group-level distributions of the relevant parameters in the relevant blocks.

were more likely than in the stable environment to choose the other slot machine than the one they chose on the last trial (within that environment) if that choice did not yield reward ($t(35)$ = 2.888, $p$ = .003).

While the dual learning rate model fitted the data better than the single learning rate model, on a group level, positive and negative learning rates do not seem to differ that much from each other. This is because, on an individual level, there is substantial variation in how much, and even in the direction in which, positive and negative learning rates differ from each other (Fig 2B).

Finally, in the control phase, where both environments had stable reward contingencies (and the slot machines were presented on different locations and had different colours than in the learning phase), the posterior probabilities that positive and negative learning rates were higher in the one environment than in the other were 0.121, and 0.716, respectively, suggesting that learning rates did not differ in this phase (Fig 2C). The posterior probability that inverse temperatures were lower in the one environment than in the other in the control phase was 0.326.

## Experiment 2

The results from Experiment 1 suggest that, in line with our hypothesis, participants used higher learning rates in the volatile than in the stable environment, even while randomly alternating between both environments on a trial-by-trial basis. To confirm that these observed differences were reliable, we aimed to replicate our findings from the learning phase of Experiment 1 with more statistical power, both in terms of participants as well as trials per participant. As such, Experiment 2 (n = 51; https://osf.io/ch74v) featured a design similar to the learning phase of Experiment 1, but consisted of four blocks of 200 trials each. Similar to Experiment 1, in each block, a new pair of slot machines was introduced in each casino.

A second aim of Experiment 2 was to study the evolution of learning rates over time. If participants really learned to associate different learning rates to the two environments, they should become better at using learning rates adapted to each environment over time (i.e., blocks). In other words, we expected to see the differences in learning rates between the two environments grow over blocks. In contrast, if the environment-specific learning rates observed in Experiment 1 merely reflected a response to differences in experienced prediction errors in each environment, the differences in learning rates should remain stable over blocks.

Replicating Experiment 1, the environment-specific dual learning rate model fitted the data best according to the LOOIC (Table 2). Also confirming our findings from Experiment 1, the posterior probabilities that positive and negative learning rates were higher in the volatile than in the stable environment were 0.998 and 0.995, respectively (Figs 3A–3B). The posterior probability that inverse temperatures were lower in the volatile than in the stable environment was 0.927. Confirming these results, in the volatile environment participants were more likely than in the stable environment to choose the other slot machine than the one they chose on the last trial (within that environment) if that choice did not yield reward ($t(50) = 3.755$, $p < .001$).

Next, we evaluated whether the differences in learning rates increased over blocks (Fig 3B). The posterior probability that positive learning rates were higher in the volatile than in the stable environment was 0.798 in the first block, 0.889 in the second block, 0.824 in the third

**Table 2. Model comparison.**

| Model | LOOIC | SE | ΔLOOIC | ΔSE |
|---|---|---|---|---|
| Environment-specific dual learning rate model | -13951 | 850 | 0 | 0 |
| Environment-specific single learning rate model | -14051 | 851 | 100 | 32 |
| Non-environment-specific dual learning rate model | -14214 | 838 | 263 | 57 |
| Non-environment-specific variable learning rate model | -14277 | 840 | 327 | 60 |
| Non-environment-specific single learning rate model | -14295 | 839 | 344 | 64 |
| Environment-specific variable learning rate model | -14390 | 840 | 439 | 96 |

*Note*. Models are ranked in descending order according to how well they fit the data. LOOIC refers to a model's approximated expected log pointwise predictive density. Higher values indicate higher out-of-sample predictive fit. SE refers to the standard error of a model's LOOIC. ΔLOOIC refers to the difference between a model's LOOIC and the top ranked model's LOOIC. ΔSE refers to the standard error of the difference between a model's LOOIC and the top ranked model's LOOIC.

block, and 0.957 in the fourth block; and the posterior probability that negative learning rates were higher in the volatile than in the stable environment was 0.807 in the first block, 0.886 in the second block, 0.997 in the third block, and 0.999 in the fourth block. However, an interaction analysis between environment (stable vs. volatile) and block (first vs. last) on learning rate (averaged over positive and negative learning rates), showed a posterior probability of 0.888, suggesting we cannot make strong conclusions for, or against, the hypothesis that learning rates changed across blocks. However, in line with our reward rate simulations (Fig 3B), which suggest that using higher learning rates in the volatile than in the stable environment yields higher rewards, reward rates were significantly higher in the fourth compared to the first block ($t(50) = 2.239$, $p = .015$).

## Experiment 3

Similar to Experiment 1, and in line with our hypothesis, the results from Experiment 2 suggest that participants learned to use higher learning rates in the volatile than in the stable environment, even while randomly switching back and forth between both environments on a trial-by-trial basis. This suggests that higher-order model parameters like learning rate can be learned in an environment-specific manner. Moreover, our model comparisons (in Experiments 1 and 2) as well as the evolution of the effect over time (in Experiment 2) suggest that the observed differences in learning rates do not merely reflect a response to differences in experienced prediction errors between the environment, but rather show that participants learned durable environment-learning rate associations. An interesting additional test of this hypothesis would be to show that participants still show a signature of these environment-specific learning rates in a subsequent phase where both environments suddenly have intermediate volatilities. Therefore, in Experiment 3, we aimed to further test whether participants did in fact learn environment-learning rate associations by testing whether the differences in learning rates persisted when the differences in volatilities disappeared but environmental cues remained the same.

Experiment 3 (n = 129; https://osf.io/kgsrv) started with a learning phase consisting of three blocks of 200 trials each, using the same design as Experiment 2 (in each block, introducing a new pair of slot machines in each casino). Crucially, however, the learning phase was followed by an unannounced test phase consisting of one block of 200 trials during which both casinos had the same intermediate level of volatility compared to the stable and volatile casinos from the learning phase. That is, in each casino, the slot machine most likely to yield reward (i.e., 85% vs. 15%) switched twice (i.e., every 29–37 trials within that casino).

Confirming the results of Experiments 1 and 2, in the learning phase, the environment-specific dual learning rate model again fitted the data best according to the LOOIC (Table 3). Also replicating our findings from Experiments 1 and 2, the analyses of the learning phase again showed posterior probabilities of 0.999 that positive and negative learning rates were higher in the volatile than in the stable environment (Figs 4A–4B). The posterior probability that inverse temperatures were lower in the volatile than in the stable environment in the learning phase was 0.929. Confirming these results, in the volatile environment participants were more likely than in the stable environment to choose the other slot machine than the one they chose on the last trial (within that environment) if that choice did not yield reward ($t(128) = 3.995$, $p < .001$).

Similar to Experiment 2, the differences in learning rates again seemed to increase over blocks (in the learning phase; Fig 4C). Specifically, the posterior probability that positive learning rates were higher in the volatile than in the stable environment was 0.925 in the first block, 0.993 in the second block, and 0.999 in the third block; and the posterior probability that

**Table 3. Learning phase model comparison.**

| Model | LOOIC | SE | ΔLOOIC | ΔSE |
|---|---|---|---|---|
| Environment-specific dual learning rate model | -28695 | 904 | 0 | 0 |
| Environment-specific variable learning rate model | -28847 | 903 | 152 | 50 |
| Environment-specific single learning rate model | -28864 | 903 | 169 | 32 |
| Non-environment-specific dual learning rate model | -29134 | 908 | 439 | 64 |
| Non-environment-specific single learning rate model | -29247 | 907 | 552 | 72 |
| Non-environment-specific variable learning rate model | -29525 | 890 | 830 | 115 |

*Note.* Models are ranked in descending order according to how well they fit the data. LOOIC refers to a model's approximated expected log pointwise predictive density. Higher values indicate higher out-of-sample predictive fit. SE refers to the standard error of a model's LOOIC. ΔLOOIC refers to the difference between a model's LOOIC and the top ranked model's LOOIC. ΔSE refers to the standard error of the difference between a model's LOOIC and the top ranked model's LOOIC.

negative learning rates were higher in the volatile than in the stable environment was 0.844 in the first block, 0.982 in the second block, and 0.997 in the third block. Now, the interaction effect between environment (stable vs. volatile) and block (first vs. last) on learning rate (averaged over positive and negative learning rates) also showed a more conclusive effect, with a posterior probability of 0.98. Moreover, in line with our reward rate simulations (Fig 4B), reward rates were significantly higher in the third compared to the first block ($t(128) = 4.721$, $p < .001$).

Importantly, in the test phase, where both environments had the same volatility, the environment-specific dual learning rate model still fitted the data best according to the LOOIC (Table 4). The posterior probabilities that positive and negative learning rates were higher in the formerly volatile than in the formerly stable environment were 0.312 and 0.905, respectively (Fig 4D). Thus, while there was a trend towards a difference in negative learning rates, hinting at a remaining difference in environment-specific learning strategies, there was no such difference in positive learning rates. However, given the substantial length of the test phase (i.e., 200 trials), it is possible that participants were more adaptive than anticipated and re-adjusted their learning strategies over the course of the test phase. Therefore, we performed two post-hoc analyses zooming in on the first half of the test phase. The posterior probability that inverse temperatures were lower in the formerly volatile than in the formerly stable environment in the test phase was 0.206.

Firstly, we fitted the environment-specific dual learning rate model to all test phase data up to the last trial before the second (and last) reward contingency switch in both environments. According to this model, the posterior probabilities that positive and negative learning rates were higher in the formerly volatile than in the formerly stable environment was 0.263 and 0.95, respectively (Fig 5A).

Secondly, we also tested whether the proportion of participants who chose the correct slot machine (i.e., the slot machine with the highest likelihood of yielding reward), was significantly different in both environments on the first trials following the first reward contingency switch in the test phase [24]. Since this analysis is explorative, we did not correct for multiple comparisons. As can be seen in Fig 5B, participants indeed showed a numerical trend to switch faster in the formerly volatile versus formerly stable casino, a difference that turned out to be significant in the third ($\chi^2 = 4.551$, $p = .033$) and fourth ($\chi^2 = 4.01$, $p = .045$) trial after the first reward contingency switch.

Both post-hoc analyses suggest that participants indeed started the test phase with different environment-specific (negative) learning rates, despite both environments already being identical in terms of volatility, but, over the course of the test phase, updated their learning strategies in accordance with the changed statistics of the environments.

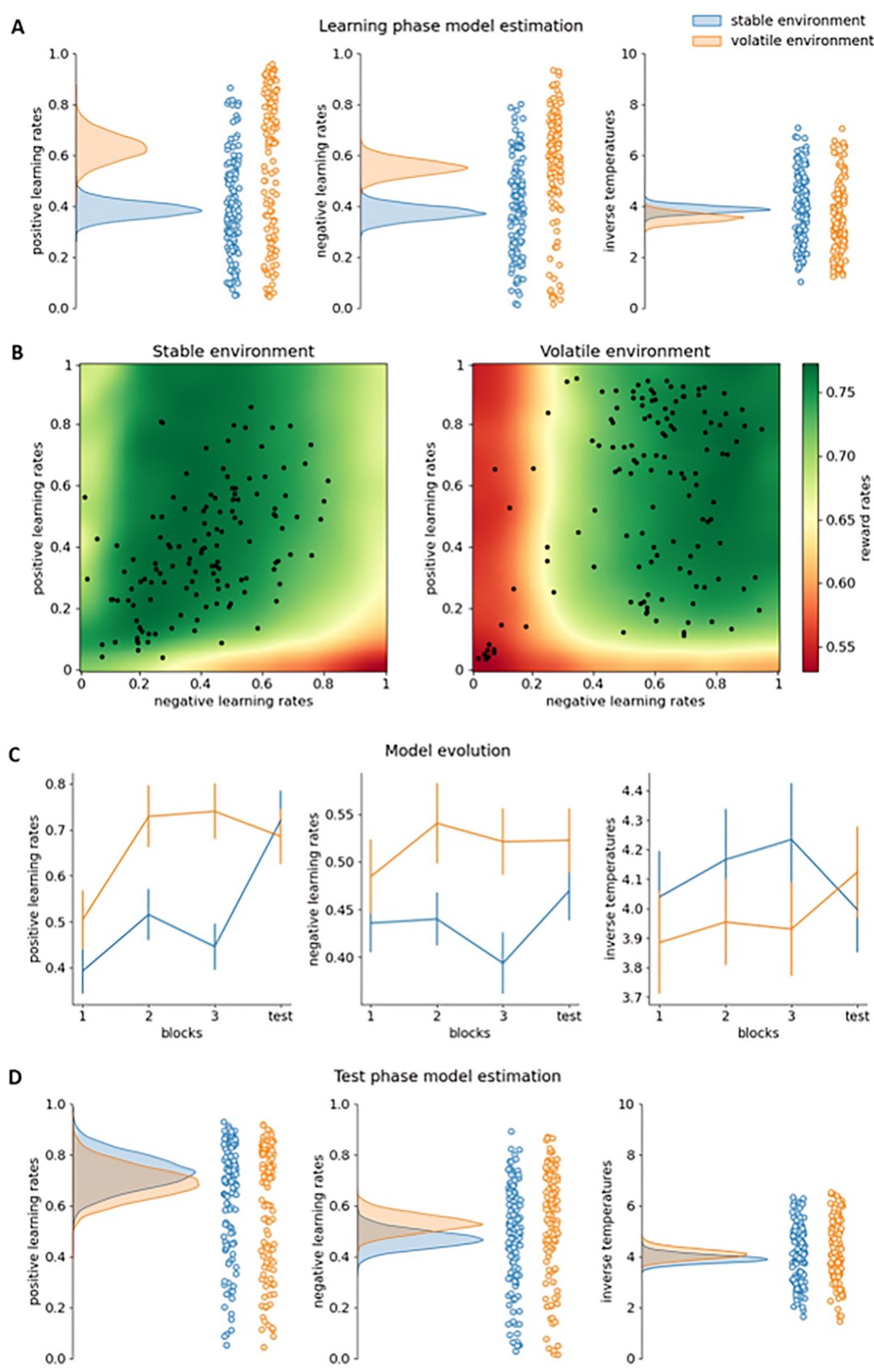

**Fig 4. Results Experiment 3. A:** Model estimation results for the learning phase. The density plots on the left side of each subfigure show the full posterior densities over the means of the group-level distributions of the relevant parameters. The scatter plots on the right side of each subfigure show the means of all individual-level posterior distributions of the relevant parameters. **B:** Reward rate simulation results for the dual learning rate model. Each point represents the (smoothed) reward rate obtained by the simulated dual learning rate model in the relevant environment and with the relevant parameter settings. Each black dot represents a participant's estimated positive and negative learning rate. **C:** Evolution of model parameters over blocks. Each subfigure shows the means and standard deviations of the posterior densities over the means of the group-level distributions of the relevant parameters in the relevant blocks. **D:** Model estimation results for the test phase.

## Discussion

The present study evaluated whether people can learn to use different environment-specific learning rates when switching back and forth between different environments, and investigate whether they effectively learned to associate these learning rates to their respective environments. To this end, we let participants randomly switch back and forth, on a trial-by-trial basis, between two different casinos. Crucially, in each casino they were faced with a different reward learning task, and each casino's reward contingencies had a different level of volatility, requiring different learning rates for optimal task performance. Across three experiments, we showed that a model with environment-specific learning rates fitted participants' data best. Specifically, participants' behaviour was always best described by the usage of higher learning rates in the volatile than the stable environment. Our findings go beyond previous studies that documented changes in learning rates over time [1,11–14], by showing that people can also switch back and forth between different learning rates across environments. This is a critical point, because it demonstrates that people can meta-learn about learning rates, and associate them to different environments.

Another important question of our study was whether people also learned to associate these learning rates to their respective environments. In contrast, it is possible that the estimated (differences in) learning rates were a by-product of experiencing more prediction errors in the volatile than in the stable environment [2,16,25,26]. To test this, we first evaluated the performance of models with fixed environment-specific learning rates against that of a model with variable non-environment-specific learning rates that varied as a function of recently experienced prediction errors. Importantly, the fixed environment-specific learning rate models always fitted the data better than the variable non-environment-specific learning rate model across all three experiments. This again suggests that participants learned to associate different learning rates to different environments.

As a second test of this hypothesis, we also expected that the differences in learning rates between environments would increase across blocks, as participants were presented with two

**Table 4. Test phase model comparison.**

| Model | LOOIC | SE | ΔLOOIC | ΔSE |
|---|---|---|---|---|
| Environment-specific dual learning rate model | -8874 | 344 | 0 | 0 |
| Environment-specific variable learning rate model | -8941 | 345 | 67 | 38 |
| Non-environment-specific dual learning rate model | -8943 | 347 | 69 | 30 |
| Environment-specific single learning rate model | -8964 | 356 | 91 | 20 |
| Non-environment-specific single learning rate model | -9012 | 351 | 138 | 37 |
| Non-environment-specific variable learning rate model | -9033 | 347 | 159 | 51 |

*Note*. Models are ranked in descending order according to how well they fit the data. LOOIC refers to a model's approximated expected log pointwise predictive density. Higher values indicate higher out-of-sample predictive fit. SE refers to the standard error of a model's LOOIC. ΔLOOIC refers to the difference between a model's LOOIC and the top ranked model's LOOIC. ΔSE refers to the standard error of the difference between a model's LOOIC and the top ranked model's LOOIC.

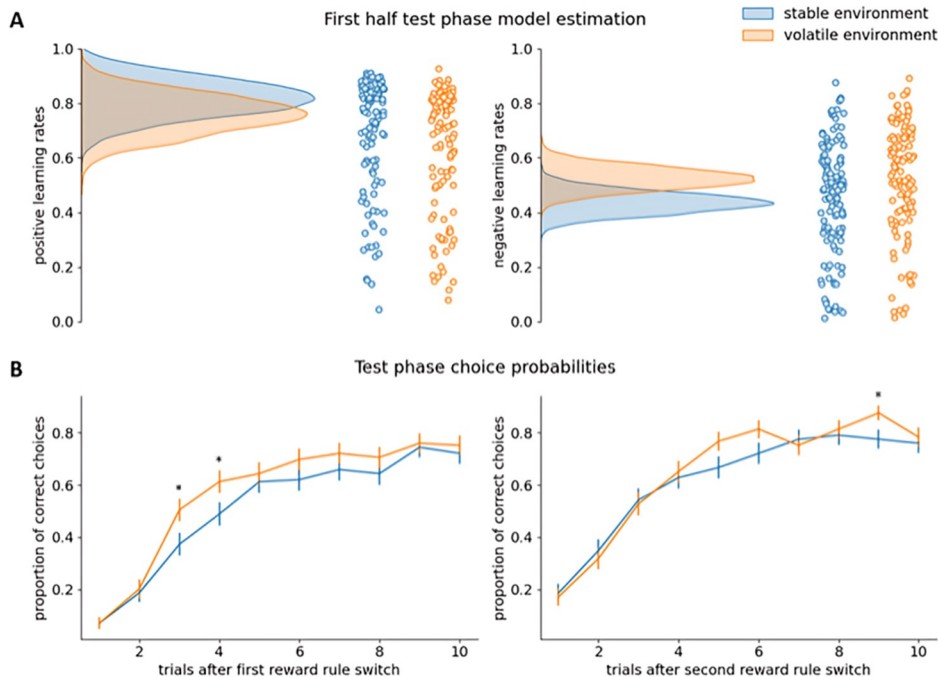

**Fig 5. Results post-hoc analyses Experiment 3. A:** Model estimation results for the test phase up to the second (and last) reward contingency switch. The density plots on the left side of each subfigure show the full posterior densities over the means of the group-level distributions of the relevant parameters. The scatter plots on the right side of each subfigure show the means of all individual-level posterior distributions of the relevant parameters. **B:** Choice probability results. Presented here are the proportion of participants choosing the correct slot machine on the first 10 trials after each of the two reward contingency switches in both environments in the test phase. Error bars represent standard errors of the choice probabilities.

new two-armed bandits in each block. Specifically, if the difference in learning rate resulted from interactions with the specific two-armed bandit, and participants failed to learn the association with the casinos, they would have to re-learn this difference in volatility for each block. Instead, we observed an increasing difference between the two environment-specific learning rates over blocks, again suggesting that participants gradually learned to optimise and associate different learning rates to both environments. On top of this, however, participants seemed to have a tendency to start learning with rather low learning rates which they then increased across blocks. This is merely conjecture, but people might have a default learning rate with which they start learning in a novel environment before they have had enough experience to adapt to it. Perhaps this default learning rate is rather low because our everyday world tends to be rather stable. Alternatively, this could also be due to the framing of our experiment in terms of gambling in a casino, where reward distributions are often thought to be extremely noisy and impossible to learn.

In Experiment 3, we further investigated this hypothesis by using a test phase where contextual cues (i.e., casino pictures and locations and slot machine colours and locations) remained the same but the differences in reward contingency volatilities disappeared. Our main analyses suggested that this mostly led to similar learning rates between the two casinos. However, this does not necessarily argue against the idea that participants formed environment-learning rate associations. First and foremost, it evidences the quick, adaptive nature of human behaviour. It might be that participants also quickly adapted to the test phase, thus abolishing possible differences in learning rate between the environments. Moreover, closer inspection of this test

phase still hinted at a signature of learned differences in learning rates. Firstly, choice probability results indicated that after the first reward contingency switch, participants were faster to switch to the other slot machine in the formerly volatile than in the formerly stable environment, indicating that they used higher (negative) learning rates there. Secondly, model fitting results of the first half of the test phase only indicated that participants did still use higher negative learning rates in the formerly volatile than in the formerly stable environment. Thirdly, also model fitting results based on the entire test phase showed a trend towards higher negative learning rates in the formerly volatile compared to the formerly stable environment. Interestingly, reward rate simulations with the dual learning rate model also indicated that, when environments differ in terms of volatility, environment-specific negative learning rates are more adaptive than environment-specific positive learning rates (Fig 2).

Although we were agnostic as to whether or not the dual learning rate models (with separate learning rates for value updating after positive versus negative prediction errors) would outperform the single learning rate models, our results clearly showed this to be the case. This is consistent with several previous studies showing that people seem to weigh unexpected positive versus negative outcomes separately when updating action values [20–24]. Perhaps most noteworthy, this is also in line with another recent study that investigated inertia of used learning rates into a separate subsequent transfer phase [24]. Specifically, they manipulated volatilities between subjects (i.e., each subject was given one volatility level) in a Wisconsin Card Sorting Task (WCST). They studied whether people would adopt the same learning rate in an unannounced test phase as in an earlier training phase where the same task, or a slightly different WCST was used. They observed inertia in the learning rates in all three experiments, but only consistently so for negative learning rates. Similarly, our test phase in Experiment 3 only hinted at transfer of the environment-specific negative learning rates, but not the positive learning rates.

Another recent and relevant study appears in [28]. These authors demonstrated that participants who were overtrained on a task (but not participants who were just criterion-trained), afterwards developed habitual responding more quickly for different stimuli, suggesting that what the participants had learned was not just stimulus-action associations, but setting the learning rates in an optimal manner. Together with the current study, these studies provide concrete evidence for the notion that people can learn not only about what are the most adaptive responses to specific stimuli in specific environments (i.e., lower-level model parameters such as response values $Q^a$), but also about what are the most effective ways to learn adaptive responses, that is, about higher-level model parameters such as learning rate. Thus, we provide support for recent theories of meta-learning [6,8,29,30], as well as recent theories of cognitive control [31–33], which posit that cognitive control is implemented in the brain as the environment-specific regulation of task execution parameters, such as learning rate.

A recent theme in computational neuroscience and artificial intelligence is that (human or artificial) agents (should) learn to cluster the environments they are confronted with; and associate different (low- or high-level) parameters to each such environment [34–36]. This approach leads to efficient learning, not least because it shields against catastrophic interference. Here, we used just two environments, so the clustering was trivial in our case. However, future studies should investigate if learning rates can also be learned for different (non-trivial) clusters of environments.

Future studies could also leverage the experimental design developed for the present study to investigate the neural underpinnings of learning rate representations and, more specifically, how they can be triggered by environment-specific cues. Our experimental design may also provide new ways to test theories about the relation between (deficits in) meta-learning and psychological pathologies. Recent theories of attention-deficit/hyperactivity disorder (ADHD),

for example, suggest that ADHD is related to impairments in the processing of reward signals [37,38]. Meanwhile recent theories of autism, for example, posit that autism is related to deficits in the detection of contextual differences in learning opportunities [13,39].

## Methods

### Ethics statement

The experiments were approved by the Ghent University Psychology and Educational Sciences Ethical Committee and participants signed informed consents prior to participation.

### Participants

All three experiments were run online. Participants were recruited through Prolific (https://www.prolific.co/). All participants were between 18 and 40 years old. 50 participants took part in Experiment 1, 63 in Experiment 2, and 160 in Experiment 3. In Experiment 3, this was double the number of participants we preregistered because after having recruited the number of participants we preregistered, the results of our main analysis (i.e., test phase model estimation results) were inconclusive (the posterior probability of a difference in negative learning rates between environments was, much like in the final analysis, close to 0.9). Therefore, we ran the experiment a second time, with the same sample size, to see whether the results would be different. The results were highly similar across both analyses, so we decided to report the results together.

In all three experiments, participants were excluded from the analyses if their reward rates were not significantly higher than chance level in either (or both) of the two environments in the learning phase (according to chi-square tests with an alfa-level of 0.05). This led to the removal of 14 participants in Experiment 1, 12 participants in Experiment 2, and 31 participants in Experiment 3. All participants completed all experimental blocks. A response time deadline of 5 seconds was implemented in all three experiments, but none of the included participants failed to give a response in more than 4.58% of trials, while 93.04% of the included participants failed to give a response in less than 1% of trials. The mean response time was 677 ms (SD = 411 ms).

All three experiments took participants about 45 minutes to complete, in return for which participants received a participation fee of £6 and an average reward bonus of £4 (in Experiment 1) or £4.5 (Experiments 2 and 3).

### Experimental design

**Experiment 1.** The experiment was programmed in jsPsych [40]. Participants performed two days (i.e., blocks) of gambling in to Las Vegas, USA (or Macau, China) (learning phase), and then one more day of gambling in Macau, China (or Las Vegas, USA) (control phase) (with location order counterbalanced across participants). In the learning phase, each trial started with the presentation of a picture of one of the two casinos, randomly selected on each trial, for 750 ms (Fig 1A). One casino was always presented on the upper half of the screen, while the other casino was always presented on the lower half of the screen. Subsequently, two slot machines were presented side by side in the same location as the casino. The slot machines were blue when presented on the upper half of the screen, and grey when presented on the lower half of the screen. The slot machines remained on the screen until a choice had been made, or until five seconds had passed. Participants were instructed that on each trial, one of the two slot machines would yield a reward, while the other one would not. After their choice,

the slot machines would remain on screen for another 750 ms, during which the lights of the chosen slot machine would flicker if that was the winning slot machine.

In the learning phase, participants performed two blocks of 240 trials, 120 trials in each casino. Crucially, and not instructed to the participants, the reward schedules were different in the two casinos. In the stable casino, one slot machine had a 75% probability of yielding a reward, while the other slot machine had a 25% probability of yielding reward, and these reward probabilities remained stable for the duration of each block. In the volatile casino, one slot machine had a 90% probability of yielding reward, while the other slot machine had a 10% probability of yielding reward (to balance the two environments in terms of difficulty, Fig 1B), but these probabilities switched every 16–24 trials (within this casino). Which casino was the stable one and which casino was the volatile one was counterbalanced across participants. The different slot machines were made recognisable by means of a letter printed on top of them. Their (left-right) location was randomly determined on each trial, and participants had to indicate their choice by providing a left or right response by pressing the f- or j-key on the keyboard, respectively. In each block and each casino, new slot machines (i.e., new letters) were used.

To make sure that participants learned that some slot machines were more rewarding than others, they were explicitly instructed that some slot machines might be more likely than others to yield reward and they had a practice phase in London before virtually travelling to Las Vegas (or Macau). This practice phase consisted of 20 trials and featured only one casino. This casino and its slot machines were presented in the centre of the screen and the slot machines were black and white. To highlight that the behaviour of the slot machines was not random, one of the two slot machines yielded a reward in 18 of the 20 trials.

After the learning phase, we also included a control phase, in which participants were introduced to two new casinos that were both stable, in case we did not observe a difference in participants' learning rates between the two casinos in the learning phase. In this case, we could study whether their learning rates were different in the learning phase compared to the control phase. This would indicate that participants did in fact adapt their decision-making strategies to the volatility of the environment, because the learning phase was overall more volatile than the control phase, but without differentiating between the two casinos (in the learning phase).

In the control phase, participants also randomly alternated between two casinos on a trial-by-trial basis. However, there were two crucial differences with the learning phase. Firstly, participants were now in a new city, and one casino was always presented on the left of the screen and had orange slot machines; while the other casino was always presented on the right of the screen and had green slot machines. Secondly, both casinos had stable reward contingencies. That is, in both casinos, on each trial, one slot machine had 75% probability of yielding reward, while the other one had only 25% probability of yielding reward, and these reward contingencies did not change over the course of the control phase.

To motivate participants, the following extra reward schedule was set up. At the end of the experiment, one bet (i.e., trial) from each of the three days (i.e., blocks) of gambling was selected at random and for each selected bet where participants had obtained reward, participants were rewarded £2 on top of their participation fee.

**Experiment 2.**  Experiment 2 used a design similar to the learning phase of Experiment 1, with two notable differences. Firstly, Experiment 2 consisted of four blocks of 200 trials each (Fig 1A). Secondly, in Experiment 1, reward rates were slightly lower in the stable compared to the volatile casino. Therefore, in Experiment 2, in the stable casino, one slot machine had 80% probability of yielding reward, while the other slot machine had 20% probability of yielding reward. Also, participants were now rewarded £1.50 on top of their participation fee for each of now four randomly selected won bets at the end of the experiment.

**Experiment 3.** Experiment 3 consisted of a learning and a test phase (Fig 1A). The learning phase consisted of three blocks of 200 trials and featured a design identical to that of Experiment 2. The test phase, which chronologically followed the learning phase, consisted of one block in which, while visually nothing changed, the two casinos became identical in terms of reward contingency volatility. That is, in the test phase, in both casinos, the reward rule switched twice (every 29–37 trials within each casino) and the currently most likely to yield reward slot machine had 85% probability of yielding reward, while the other slot machine had 15% probability of yielding reward.

## Model estimation and selection

In all three experiments, we fitted six models to the data. The first two models were the environment-specific and the non-environment-specific versions of the Rescorla-Wagner (RW) model [18]. According to the RW model, participants learn the values of different slot machines using the delta learning rule:

$$Q_{t+1}^a = Q_t^a + \alpha(r_t - Q_t^a)$$

where $\alpha$ is the learning rate, which takes a value between 0 and 1 and quantifies the extent to which $Q^a$ is updated by the reward prediction error $(r_t - Q_t^a)$. Participants then choose slot machines, according to the RW model, using the softmax decision rule:

$$p_t^a = \frac{exp\ (\beta\ Q_t^a)}{exp\ (\beta\ Q_t^a) + exp\ (\beta\ Q_t^b)}$$

where $\beta > 0$ is the inverse temperature, which quantifies the level of exploration, with higher values indicating a lower level of exploration. $\beta$ was bounded between 0 and 10 and $Q^a$ and $Q^b$ were initialised at 0. In the environment-specific version of the model, each participant was assumed to use separate learning rates and inverse temperatures for each environment, while in the non-environment-specific version of the model, participants were assumed to use the same learning rate and inverse temperature in both environments.

The next two models were the environment-specific and non-environment specific versions of an extension of the RW model, which uses separate learning rates after positive and negative prediction errors (i.e., after rewarded and unrewarded trials, respectively), henceforth referred to as the *dual learning rate model* [20–24]. These models are identical to the environment-specific and non-environment specific versions of the RW model, respectively, except that the delta learning rule now takes the form:

$$Q_{t+1}^a = \begin{cases} Q_t^a + \alpha^+\ (r_t - Q_t^a)\ if\ r_t = 1 \\ Q_t^a + \alpha^-\ (r_t - Q_t^a)\ if\ r_t = 0 \end{cases}$$

where $\alpha^+$ and $\alpha^-$ are the positive and negative learning rates, respectively. The dual learning rate model was not yet mentioned in the preregistration for Experiment 1 because we had not yet considered it, but since it fit the data of Experiment 1 better than the single learning rate model, it became our main focus in the preregistrations for both Experiments 2 and 3.

The last two models we fitted were the environment-specific and non-environment-specific versions of another extension of the RW model, which uses a variable learning rate that depends on recently experienced prediction errors, henceforth referred to as the *variable learning rate model* [2,16,25,26]. These models are identical to the environment-specific and non-environment specific versions of the RW model, respectively, except that here the relevant

learning rate is updated after each trial in which it was used according to:

$$\alpha_{t+1} = \eta |r_t - Q_t^a| + (1 - \eta)\alpha_t$$

where $\eta$ is a higher-order learning rate, which also takes a value between 0 and 1 and determines how much learning rates are affected by reward prediction errors. The variable learning rate model was not mentioned in any of the preregistrations. Nevertheless, we decided to include it to evaluate an alternative interpretation of why we observe different learning rates in the volatile compared to the stable environments of all three experiments. That is, (the non-environment-specific version of) this model postulates that different learning rates emerge because participants respond to (locally) experienced prediction errors rather than actually keeping track of environment-specific learning rates. Hence, if this is indeed the reason for differences in learning rates between environments, this variable learning rate model should fit the data better than the other (fixed learning rate) models. An overview of all six model can be found in Table 5.

We used hierarchical Bayesian analysis (HBA) to fit these models to the data. HBA has two main advantages compared to maximum likelihood estimation (MLE). Firstly, by means of Markov Chain Monte Carlo (MCMC) sampling, Bayesian modelling allows for the estimation of posterior probability distributions over free model parameters, thus quantifying uncertainty in parameter estimates, rather than merely providing point estimates. Secondly, MLE assumes that data across participants are statistically independent, ignoring similarities among participants, resulting in sub-optimal model estimation. Instead, HBA maximally leverages all available data by simultaneously estimating participant- and group-level parameters, which bidirectionally constrain each other. Previous studies also indicated that HBA results in more reliable parameter estimates than MLE [41].

The HBA was performed in Stan [42], which uses Hamiltonian Monte Carlo (HMC) sampling, a variant of MCMC sampling. For each combination of phase (learning vs. control/test) and environment (stable vs. volatile), individual-level free parameters were assumed to be drawn from a group-level normal distribution specific to that combination of phase and environment. For the means of these group-level distributions, we used uniform priors between the lower and upper bounds of the relevant parameter, while for the standard deviations we used half-Cauchy (0, 5) priors. For the individual-level parameters we also used bounded uniform priors. To minimize the dependence between the means and standard deviations of group-level distributions, we used non-centred parameterisations. To maximise the efficiency

**Table 5. Model overview.**

| Model | $\alpha$ | $\alpha^+$ | $\alpha^-$ | $\eta$ | $\beta$ |
|---|---|---|---|---|---|
| Environment-specific single learning rate model | 2 | 0 | 0 | 0 | 2 |
| Non-environment-specific single learning rate model | 1 | 0 | 0 | 0 | 1 |
| Environment-specific dual learning rate model | 0 | 2 | 2 | 0 | 2 |
| Non-environment-specific dual learning rate model | 0 | 1 | 1 | 0 | 1 |
| Environment-specific variable learning rate model | 2 | 0 | 0 | 2 | 2 |
| Non-environment-specific variable learning rate model | 1 | 0 | 0 | 1 | 1 |

*Note.* $\alpha$ is the learning rate; $\alpha^+$ and $\alpha^-$ are the positive and negative learning rates, respectively; $\eta$ is the higher-order learning rate; and $\beta$ is the inverse temperature. 0 indicates that the model (row) does not contain the parameter (column); 1 indicates that the model contains the parameter once, shared across environments; and 2 indicates that the model contains the parameter twice, once for each environment specifically.

of HMC sampling, parameters were first estimated in an unbounded space and then probit-transformed to the relevant bounded space [19].

For each model, 4000 samples were drawn from the posterior distributions, the first 1000 of which were discarded as burn-in, across four sampling chains, resulting in a total of 12,000 posterior samples. Convergence of posterior distributions was checked by visually inspecting the traces and by numerically checking the Gelman-Rubin statistics [43], which were all well below 1.1, for each estimated parameter.

We used the leave-one-out information criterion (LOOIC) [27] for model comparisons. The LOOIC can be computed from the log pointwise posterior predictive density of observed data and provides an estimate of the out-of-sample predictive accuracy of a model [27]. A more complex model (i.e., a model with more free parameters) can be considered superior to a less complex model when the LOOIC of the former is higher than that of the latter and this difference is larger than two times the standard error of this difference. In the preregistration for Experiment 1 we stated we would use the Akaike information criterion (AIC) [44] rather than the LOOIC because we had not yet decided to use hierarchical Bayesian analyses (HBA) rather than maximum likelihood estimation (MLE) to fit models to data. Nevertheless, we did use HBA, for reasons outlined above, allowing us to use the more reliable LOOIC, which uses the pointwise log-likelihood of the full Bayesian posterior distribution, rather than the AIC, which uses only point estimates, for model comparisons. The subsequent preregistrations for Experiments 2 and 3 also mentioned this.

To calculate the posterior probability that learning rates were higher in the volatile than in the stable environment, we calculated the proportion of posterior samples in which the group-level mean learning rate was higher in the volatile compared to the stable environment. Thus, a posterior probability higher than 95% corresponds to a one-tailed p-value lower than 0.05.

## Reward rate simulations

To make sure that the aforementioned design details indeed required participants to use a higher learning rate to maximise reward in the volatile than in the stable environment, we performed reward rate simulations. That is, using the aforementioned design details for the learning phase and the single learning rate model, we simulated data for each combination of learning rates {0.01, 0.02, . . ., 0.99} and inverse temperatures {0.1, 0.2, . . ., 9.9} for each environment separately and computed the reward rate for each of these simulations. These simulations indicated that, to maximise reward, participants would indeed be required to use a higher learning rate in the volatile than in the stable environment (Fig 1B). They also indicated that, if participants would use optimal parameter settings, they would obtain similar reward rates in both environments. We first performed reward rate simulations with the single learning rate model because we had not yet considered the dual learning rate model. However, when we observed that the dual learning rate model fitted the data better than the single learning rate model in Experiment 1, we repeated these simulations with the dual learning rate model, finding similar results (Fig 2).

## Model recovery simulations

To ensure that the experimental design and model selection procedure would allow for the reliable selection of the model fitting the data best, we performed model recovery simulations. Specifically, using the learning phase described above, we simulated 10 datasets with each of the six models described above. For all model parameters we used the estimated posterior means from Experiment 1 as the means of the group-level distribution. These means were 0.4 and 0.6 for stable and volatile learning rates, respectively, in all three environment-specific

**Table 6. Model recovery simulation results.**

| Model | 1 | 2 | 3 | 4 | 5 | 6 |
|---|---|---|---|---|---|---|
| Environment-specific dual learning rate model (1) | 0.9 | 0.1 | 0 | 0 | 0 | 0 |
| Environment-specific single learning rate model (2) | 0 | 1 | 0 | 0 | 0 | 0 |
| Environment-specific variable learning rate model (3) | 0 | 0 | 1 | 0 | 0 | 0 |
| Non-environment-specific dual learning rate model (4) | 0 | 0 | 0 | 1 | 0 | 0 |
| Non-environment-specific single learning rate model (5) | 0 | 0 | 0 | 0 | 1 | 0 |
| Non-environment-specific variable learning rate model (6) | 0 | 0 | 0 | 0 | 0 | 1 |

*Note.* Presented here are the probabilities that a given model (column) fitted a dataset generated with a given model (row) best.

models; 3.5 and 4.5 for stable and volatile inverse temperatures, respectively, in all three environment-specific models; 0 for the stable as well as the volatile higher-order learning rate in the variable learning rate model; 0.5 for learning rates in all non-environment-specific models; 4 for inverse temperatures in all non-environment specific models; and 0 for the higher-order learning rate in the variable learning rate model. We then randomly sampled 36 (i.e., the number of participants we used in Experiment 1) values from each group-level distribution with the relevant mean and an SD of 0.2 for learning rates in all models, 2 for inverse temperatures in all models, and 0.1 for higher-order learning rates in the variable learning rate models. For each model, we then simulated a dataset using these parameter values and two blocks of 120 trials per participant (i.e., the number of trials we used in Experiment 1) and fitted all models to this simulated dataset using the model estimation procedure described above. Finally, using the model selection procedure described above, we tested for each simulated dataset which model fitted it best. As can be seen in Table 6, all models can be reliably recovered. In experiments 2 and 3, we increased the sample size in terms of participants as well as trials per participant. Thus, here, model recovery rates were undoubtedly at least as high.

## Parameter recovery simulations

To make sure that the experimental design and model estimation procedure would allow for the reliable estimation of the environment-specific dual learning rate model parameters, we performed parameter recovery simulations. Specifically, using the aforementioned design details for the learning phase and the dual learning rate model, we simulated 64 datasets for each environment separately. We used each combination of positive and negative learning rates {0.2, 0.4, 0.6, 0.8} and inverse temperatures {2, 4, 6, 8} as the true mean of the group-level distribution of these parameters. For each combination, we then randomly sampled 36 (i.e., the number of participants we used in experiment 1) true positive and negative learning rates and inverse temperatures from the group-level normal distribution with the relevant mean and an SD of 0.2 for learning rates and 2 for inverse temperatures (i.e., the group-level posterior SDs we observed). For each combination, we then simulated a dataset using these parameter values and two blocks of 120 trials per participant (i.e., the number of trials we used in experiment 1) and fitted the dual learning rate model to this simulated dataset using the model estimation procedure described above. Finally, for each environment separately, we calculated parameter recovery rates by correlating all (64 datasets x 36 participants = 2304) true parameters to the corresponding estimated parameters, that is, the individual-level posterior means. These simulations indicated that the model could be reliably fitted to our behavioural data. Positive learning rate recovery rates were 0.824 in the stable environment and 0.856 in the volatile environment. Negative learning rate recovery rates were 0.914 in the stable environment

and 0.925 in the volatile environment. Inverse temperature recovery rates were 0.921 in the stable environment and 0.932 in the volatile environment. In experiments 2 and 3, we increased the sample size in terms of participants as well as trials per participant. Thus, here, parameter recovery rates were undoubtedly at least as high.

## Model validation simulations

For validating the model, we used the posterior predictive check method. This method takes participants' fitted model parameters and uses them to simulate choices given the trial sequence they were presented with. Simulated and true choice patterns can then be compared to determine how well the model captures participants' behaviour [45]. Specifically, for each participant, we used parameter estimates from each of their participant-level joint posterior samples and used them to simulate a choice sequence conditional on the trial sequence this participant was presented with. For each simulated choice sequence, we then calculated the obtained reward rate and averaged the reward rates over all 12,000 simulated choice sequences. Finally, we correlated participants' true reward rates with the average reward rates obtained by the simulations.

The posterior predictive check indicated that the environment-specific dual learning rate model, which fitted the data best in all three experiments, indeed adequately captured participants' behaviour. The Pearson correlation between participants' true reward rates (from the learning phase) and the average reward rates obtained by simulating data using their parameter estimates was 0.865 in the stable and 0.889 in the volatile environment in experiment 1, 0.958 in the stable and 0.883 in the volatile environment in experiment 2, and 0.86 in the stable and 0.849 in the volatile environment in experiment 3.

## Author Contributions

**Conceptualization:** Tom Verguts, Senne Braem.

**Data curation:** Jonas Simoens.

**Formal analysis:** Jonas Simoens.

**Funding acquisition:** Jonas Simoens, Tom Verguts, Senne Braem.

**Investigation:** Jonas Simoens.

**Methodology:** Jonas Simoens, Tom Verguts, Senne Braem.

**Project administration:** Jonas Simoens, Tom Verguts, Senne Braem.

**Resources:** Jonas Simoens.

**Software:** Jonas Simoens.

**Supervision:** Tom Verguts, Senne Braem.

**Validation:** Jonas Simoens.

**Visualization:** Jonas Simoens.

**Writing – original draft:** Jonas Simoens.

**Writing – review & editing:** Tom Verguts, Senne Braem.

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
