## [Decision Letter · Decision Letter 0]

6 Sep 2023

Dear Simoens,

Thank you very much for submitting your manuscript "Meta-learning environment-specific learning rates" for consideration at PLOS Computational Biology.

As with all papers reviewed by the journal, your manuscript was reviewed by members of the editorial board and by several independent reviewers. In light of the reviews (below this email), we would like to invite the resubmission of a significantly-revised version that takes into account the reviewers' comments.

We cannot make any decision about publication until we have seen the revised manuscript and your response to the reviewers' comments. Your revised manuscript is also likely to be sent to reviewers for further evaluation.

Sincerely,

Lusha Zhu, Ph.D.

Academic Editor

PLOS Computational Biology

Marieke van Vugt

Section Editor

PLOS Computational Biology

Reviewer's Responses to Questions

**Comments to the Authors:**

Reviewer #1: Review of Meta-learning environment-specific learning rates by Simoens et al.

Summary

In this study, the authors investigated the relationship between learning rates and environment volatility in reinforcement learning tasks. Across 3 different experiments and variations of dual bandit tasks with different volatility conditions, they show that reinforcement learning models implementing environment specific learning rates fitted better the data compared to models that were myopic to volatility conditions. They found that subjects used overall higher learning rates in volatile environments and were able to switch between high and low learning rates depending on the environment they interacted with.

The question investigated in the present study (i.e., Can people learn to use environment specific learning rates adapted to different volatility conditions?) is interesting and various experiments and analyses have been implemented to answer that question. However, analyses and results are not entirely convincing and do not answer satisfactorily the aforementioned question. In its current version, the manuscript also lacks clarity and details across all sections. I introduce below a series of major and minor comments.

Major Comments

#1 Model recovery

In the current version of the manuscript, the authors provide a parameter recovery analysis in the methods section. However, no model recovery analysis is provided. This analysis is of prime importance insofar as the model comparison presented in the results section constitutes the primary result of the study. This would confirm that the model selection procedure used to fit the data in this particular task is actually able to distinguish the different models. Could the authors provide such an analysis?

#2 (inverse) Temperature parameter

The temperature parameter is totally absent in the current version of the results. This parameter is an essential piece in analyses such as those presented, as a difference in Q-values has a radically different impact on choice depending on the temperature value. Considering the importance of this parameter, the results section and/or the figures should include a comparison of the temperature parameters in the two volatility conditions.

#3 Double learning rate model

Authors found that models with different learning rates for positive and negative prediction errors fit better the data. Authors state that “This is likely because negative learning rates are more important than positive learning rates when adapting to changed reward contingencies.” However they give no details about the mechanism underlying this result, and the fitted learning rates (for positive and negative prediction errors) seem equal to one another (Figure 3). Could the authors give details about these mechanisms? It would be helpful to see if a potential asymmetry in learning rates differs between the two volatility conditions and across periods (experiment 2), and could explain partly the results observed.

If there is no interaction between learning rates asymmetry and volatility conditions, it would then be interesting to see how would perform a model with only two learning rates (for positive and negative PEs) and a single scaling parameter (adjusting learning rates between the two volatility conditions) compared to the winning model presented here.

#4 Between subjects analysis

From figures 3-5, it seems that subjects have very different learning rates. How these differences relate to the current results? Is there a group that adapts to the volatility of the environment and another one that does not? This is also important in regards to the message of the manuscript where it is stated that people use “optimal” learning rates in both environments, whereas the range of learning rate values of subjects in the experiments is extremely large.

#5 Do people switch back and forth between two pairs of learning rates according to the environment they interact with?

On average, learning rates are found to be higher in the high volatility condition but there is little evidence of a switch back and forth between two pairs of “optimal” learning rates. The test phase from experiment 3 shows that subjects adapt quickly (over a block) to the volatility of the environment. Results from experiments 2 and 3 could then be interpreted as a series of adaptations to high and low volatility environments. The hypothesis of an increase in the difference between learning rates over blocks found statistical evidence in experiment 3 only. The authors state that subjects learn to use “optimal” learning rates over blocks but do not provide any measures indicating that learning rates are indeed more “reward maximising” in the last block compared to the first one.

Minor Comments

#1 “optimal” temperature for each environment

Concerning the simulations performed with the winning model, authors state that they used the “ inverse temperatures fixed to the optimal value for each environment”. It would be very helpful (even essential) for the reader’s understanding to know what are the values of these parameters and if they differ, why this is the case.

#2 Number of parameters of the winning model

Even though we can guess that the winning model has 6 parameters, 4 learning rates and 2 temperatures, it is hard to be sure of that as it is not written clearly in the manuscript. I would suggest to detail the list of parameters used for each model, summarised in a table for instance.

#3 Contingencies and volatility description

In the manuscript, there is no description of the contingencies used in stable and volatile environments nor description of the volatile environment itself. The reader needs those details to fully understand the paradigms and results. For increased clarity, I would recommend to add these details in the main text and not only in the method section.

#4 Medium volatility environments

The test phase of the third experiment compounds 200 trials (Figure 1) and reward contingencies switched twice, every 29-37 trials (Methods). Is there a typo in one of these two pieces of information? It is not clear when the switches happen along the 200 trials.

#5 “Optimal” and “belief”

Seeing Q-values in reinforcement learning as beliefs (introduction) is not usual and should be justified.

Similarly, “optimal” is used extensively in the manuscript and its use seems inappropriate in many instances. In the abstract, people are described as using “optimal” learning rates, which does not seem to be the case. In the first paragraph of the results section, we can read that an “optimal” learner would use two different learning rates in the environment described, but it is not clear what optimal means for the authors here. In general? In the specific context of the model considered?

Reviewer #2: The authors tested whether participants could adjust their learning rates according to environmental cues. Through fitting the models to empirical data from three experiments, the authors consistently found in both the model comparison and the fitted parameters that: 1) the environment-specific learning rate (LR) model was the winning model, and 2) during the learning phase, learning rates in volatile environments were higher. Based on these modeling results, the authors concluded that people could meta-learn the learning rate.

The modeling results were clear. I believe the main contribution of this paper is associating the adjustment of the learning rate with an environmental cue. In general, the paper's conclusion and results didn't surprise me, but they represented a solid forward step. In my opinion, the most valuable part of this paper is the data; therefore, I suggest the authors make their data open-access after publication. When I’m doing the review, the osf website says that I need to press a “request access” button to apply for access permission, which is not ideal.

About the “meta”

I feel that the word "meta" is overused in this paper. In the current experiment, all LR adjustments were based on a visual cue (a casino icon), which didn't seem very "meta". More importantly, the authors didn't provide a process model for the environment-specific LR adjustment. Experiments 2 & 3 showed that the LR was changing, especially in the volatile condition. However, the current model didn't provide insight into the normative theory of the LR adjustment at the process level. Removing the "meta" from the title might be better.

A related point of interest could be a discussion on how such LR adjustments are achieved by the human brain, both at the algorithm level and the neural implementation level.

About the model-free analysis

The authors used model comparison and fitted parameters as two main supports for their conclusions. While a model-based analysis is good and valuable, I wonder if it would be possible to include some model-free analysis of the data. Model-free analysis is valuable because it usually requires fewer mathematical assumptions and can illustrate behavioral phenomena more directly. Figure 6B represents a model-free analysis, but I believe it's insufficient. For example, the authors could try using "whether participants made a mistake in the last trial" as a regressor to predict "whether participants switch their decision in the future trial". After performing the regression, the authors could compare the regression coefficients in different conditions and gain more model-free intuition into the data. A similar analysis was performed in (Vickery et al., 2011).

About the optimality analysis

Given that Behrens has already conducted a related simulation, I'm not surprised that people should adjust the learning rate in different environments. For me, Figures 1B and 2 don’t seem necessary.

Moreover, I think the phrase "people can learn to use different, optimal learning rates" in the abstract is an overstatement. The current optimality analysis shows that the direction of the fitted LR is good (i.e., high LR in a volatile environment), but it's difficult to determine whether such LR is optimal. Optimality is a strong claim and should be used cautiously. To make such a claim, the authors might need a different model format (Lieder & Griffiths, 2017, 2020; Sims et al., 2012). A classic form of such a format includes an "argmax" in the model.

When I read "Here, using optimality simulations and Bayesian hierarchical modelling across three experiments, we show that people can learn to use different, optimal learning rates when switching back and forth between two different environments." in the authors' abstract, I initially thought that the authors were modeling the agent as an optimal Bayesian decision maker in a hierarchical environment, but then I found I had misunderstood it. Since "Bayesian hierarchical modeling" is just a fitting technique, removing it could help the reader avoid possible confusion.

About the Model validation

I recommend moving the model validation to the supplementary material for readability. Additionally, did the authors try other model validation methods (e.g., parameter recovery)? Recovering the reward rate doesn't provide strong evidence that the model has a valid fit to the data because the reward rate is a crude behavioral metric.

References

Lieder, F., & Griffiths, T. L. (2017). Strategy selection as rational metareasoning. Psychological Review, 124(6), 762.

Lieder, F., & Griffiths, T. L. (2020). Resource-rational analysis: Understanding human cognition as the optimal use of limited computational resources. Behavioral and Brain Sciences, 43, e1.

Sims, C. R., Jacobs, R. A., & Knill, D. C. (2012). An ideal observer analysis of visual working memory. Psychological Review, 119(4), 807–830. https://doi.org/10.1037/a0029856

Vickery, T. J., Chun, M. M., & Lee, D. (2011). Ubiquity and Specificity of Reinforcement Signals throughout the Human Brain. Neuron, 72(1), 166–177. https://doi.org/10.1016/j.neuron.2011.08.011

Reviewer #3: In their paper, Simoens and colleagues address one main question with help of three experiments: Do humans meta-learn about the volatility of reward environments and adapt their learning rates accordingly. The meta-learning is tested by randomly exposing participants to different reward environments of different volatility and see whether participants learn with different learning rates on a trial-by-trial basis. Furthermore, they investigated whether participants actually associate different environments with different learning rates (and thus apply them according to the given environment) or whether the different learning rates are just a by-product of a difference in experienced prediction errors.

The paper is well written and uses well-designed experiments and state-of-the-art hierarchical Bayesian modeling to answer the research questions and use parameter recovery validate the fitting procedure. The methods and hypotheses have been pre-registered and deviations are clearly described and motivated. All in all, the manuscript is of high quality and I do not have any major concerns, but rather some additional questions:

1) Since the experiment was performed online there is much less control over the testing environment. This is also reflected in the high number of participants that had to be excluded because their reward rate did not significantly exceed that of random choice behavior. I thus think it would be important to add some further characterization of behavior for the included participants. This might be things like reaction times or missed trials (maybe for the different blocks to see changes over time), maybe the authors have some other ideas. Also, did all included participants complete all blocks of the respective experiments?

2) Another interesting measure would be the characterization of how close to optimality participants learn. Maybe you could overlay the simulation plots in Figure 2 with points showing the estimated learning rates for each participant, color-coded in with the reward rate?

3) I am wondering why there is a general tendency to start learning with too low learning rates which then increase over time. You obviously investigate your main question, whether the difference between the learning rates in the stable vs. the volatile environment increases over time. But going beyond that, do you have any idea why the general trend for the learning rates is to increase, i.e. why do you not see participants starting with too high learning rates that decrease towards optimality? Looking at the simulation plots in Figure 2 you nicely show that only looking at the positive or the negative learning rate, it is not always possible to say whether a given learning rate is optimal or not, it is the combination of both that really matters for achieving a high reward rate. Therefore, the plots in figures 4b and 5c are difficult to interpret in relation to reward-rate optimization. Could you maybe somehow elucidate this trend more in relation to reward rate optimization?

Kind regards,

David Meder

**Have the authors made all data and (if applicable) computational code underlying the findings in their manuscript fully available?**

Reviewer #1: Yes

Reviewer #2: **No: **It's in OSF website, but it still requires application to access the data and the code.

Reviewer #3: Yes

PLOS authors have the option to publish the peer review history of their article (what does this mean?). If published, this will include your full peer review and any attached files.

Reviewer #1: No

Reviewer #2: No

Reviewer #3: **Yes: **David Meder
---

## [Decision Letter · Decision Letter 1]

9 Mar 2024

Dear Simoens,

We are pleased to inform you that your manuscript 'Learning environment-specific learning rates' has been provisionally accepted for publication in PLOS Computational Biology. Please follow Revewer 1's suggestion, clarify ""intermediate level of volatility", reward contingencies, and the number of probability reversals in Page 14. 

Best regards,

Lusha Zhu, Ph.D.

Academic Editor

PLOS Computational Biology

Marieke van Vugt

Section Editor

PLOS Computational Biology

Reviewer's Responses to Questions

**Comments to the Authors:**

Reviewer #1: I thank the authors for their detailed answers to my concerns. They provided a pertinent set of analyses and details about the methods together with an increased clarity of the manuscript in general, that improve substantially the study.

Minor comment.

Page 14, last paragraph

It would help the reader to know in that paragraph, what is the "intermediate level of volatility", what are the reward contingencies and what is the number of probability reversals.

Reviewer #2: The authors have done a great job addressing all my concerns. Compared to the previous version, which focused on meta and optimality, the current narrative is more solid, albeit less fancy. Most importantly, by making their data available to the community, the authors provide a significant benefit. I believe the current version is sufficiently good.

Reviewer #3: The authors have addressed all my concerns. I recommend publication and I'm looking forward to seeing the manuscript published.

**Have the authors made all data and (if applicable) computational code underlying the findings in their manuscript fully available?**

Reviewer #1: Yes

Reviewer #2: Yes

Reviewer #3: Yes

PLOS authors have the option to publish the peer review history of their article (what does this mean?). If published, this will include your full peer review and any attached files.

Reviewer #1: No

Reviewer #2: No

Reviewer #3: **Yes: **David Meder

---

## [Editor Report · Acceptance letter]

14 Mar 2024

PCOMPBIOL-D-23-00985R1 

Learning environment-specific learning rates

Dear Dr Simoens,

I am pleased to inform you that your manuscript has been formally accepted for publication in PLOS Computational Biology. Your manuscript is now with our production department and you will be notified of the publication date in due course.

With kind regards,

Bernadett Koltai
